# SUPERVISED RANDOM FEATURE REGRESSION VIA PROJECTION PURSUIT

## ABSTRACT

Random feature methods and neural network models are two popular nonparametric modeling methods, which are regarded as representatives of shallow learning and Neural Network, respectively. In practice random feature methods are short of the capacity of feature learning, while neural network methods lead to computationally heavy problems. This paper aims at proposing a flexible but computational efficient method for general nonparametric problems. Precisely, our proposed method is a feed-forward two-layer nonparametric estimation, and the first layer is used to learn a series of univariate basis functions for each projection variable, and then search for their optimal linear combination for each group of these learnt functions. Based on all the features derived in the first layer, the second layer attempts at learning a single index function with an unknown activation function. Our nonparametric estimation takes advantage of both random features and neural networks, and can be seen as an intermediate bridge between them.

## 1 INTRODUCTION

Kernel methods are one of the most powerful methods for nonlinear statistical learning problems attributed to their excellent statistical theories and flexible modeling framework. Using the randomized algorithms for approximating kernel matrices, random feature (RF) models attract increasing attention due to that they significantly reduce the extensive hand tuning form the user for training, but obtain similar or better prediction accuracy with limited data size compared to neural network models (Du et al., 2022; Zhen et al., 2020). The RF model can be traced back to the work of (Rahimi & Recht, 2007), and was successfully developed by (Li et al., 2019b). To be specific, for observations $(y_i, \mathbf{x}_i)_{i=1}^n, \mathbf{x}_i \in \mathbb{R}^p, y_i \in \mathbb{R}$, RF models consider a linear combination over a set of prespecified nonlinear functions on a relatively low-dimensional randomized feature space to predict $y$. That is,

$$y_i = f(\mathbf{x}_i) + \varepsilon_i := \sum_{j=1}^N \alpha_j \sigma(\langle \mathbf{x}_i, \boldsymbol{\theta}_j \rangle / \sqrt{p}) + \varepsilon_i, \quad i = 1, \cdots, n, \tag{1}$$

where $N \to \infty$, $\langle \boldsymbol{\alpha}, \mathbf{x} \rangle = \sum_{j=1}^p \alpha_j x_j$, and $\sigma(\cdot)$ is a pre-specified function, like Relu or the Sigmoid function. Here, $\boldsymbol{\theta}_j$ is chosen randomly from a prespecified distribution, say, a unit ball, i.e., $\boldsymbol{\theta}_j \sim \mathrm{Unif}(\mathbb{S}^{p-1}(\sqrt{p}))$, where $\mathbb{S}^{(d-1)}(r)$ denotes the sphere of radius $r$ in $d$ dimensions, and $r = \sqrt{d}$. Model equation 1 involves unknown parameters $\alpha_j, j = 1, \cdots, N$ only. The coefficients $\boldsymbol{\alpha}$ in the RF model can be estimated using the following ridge regression:

$$\hat{\boldsymbol{\alpha}}(\lambda) = \arg \min_{\alpha \in \mathbb{R}^N} \left\{ \frac{1}{n} \sum_{i=1}^n \left( y_i - \sum_{j=1}^N \alpha_j \sigma(\langle \boldsymbol{\theta}_j, \mathbf{x}_i \rangle) \right)^2 + \frac{N\lambda}{p} \|\boldsymbol{\alpha}\|_2^2 \right\}. \tag{2}$$

Let $\mathcal{F}_{RF}(\Theta) = \left\{ f(\mathbf{x}) = \sum_{i=1}^N \alpha_i \sigma(\langle \boldsymbol{\theta}_i, \mathbf{x} \rangle) : \alpha_i \in \mathbb{R} \, \forall i \leq N \right\}$, where $\Theta \in \mathbb{R}^{N \times p}$ is a matrix whose $i$-th row is the vector $\boldsymbol{\theta}_i$. When the number of random features, $N$, goes to infinity, under a suitable bound on the $\ell_2$ norm of the coefficients, $\mathcal{F}_{RF}$ reduce to certain Reproducing Kernel Hilbert Space (RKHS) (Liu et al., 2020). Specifically, the ridge regression over the function class converges to kernel ridge regression (KRR) with respect to the kernel: $H_p^{RF}(\mathbf{x}_1, \mathbf{x}_2) := h_p^{RF}(\langle \mathbf{x}_1, \mathbf{x}_2 \rangle_p) = \mathbb{E}\left[\sigma(\langle \boldsymbol{\theta}, \mathbf{x}_1 \rangle)\sigma(\langle \boldsymbol{\theta}, \mathbf{x}_2 \rangle)\right]$. Here, the expectation is with respect to $\boldsymbol{\theta}$. Clearly, distinct distributions generating $\boldsymbol{\theta}_j$ and different activation functions induce different RKHS spaces. For examples, when $\boldsymbol{\theta}$ follows a standard multivariate normal distribution, and the activation function is the ReLU activation $\sigma(x) = max(0, x)$, the kernel corresponds to the first order arc-cosine kernel. Another example

is, if the activation function is $\sigma(x) = [cos(x), sin(x)]^\top$, the kernel corresponds to the Gaussian kernel (Rahimi & Recht, 2007; Liu et al., 2020). According to Bochner's theorem, the spectral distribution $\mu_k$ of a stationary kernel $k$ is the finite measure induced by a Fourier transform, i.e., $k(\boldsymbol{x} - \boldsymbol{x}') = \int \exp\left(i\boldsymbol{\theta}^\top(\boldsymbol{x} - \boldsymbol{x}')\right)\mu_k(d\boldsymbol{\theta})$. However, it is known that the distribution and the activation function may meet misspecification issues on the function space leading to inefficient or even wrong estimation(Sinha & Duchi, 2016; Derakhshani et al., 2021).

Note that general kernel $k(\boldsymbol{x}, \boldsymbol{x}')$ describes the distance $\|\boldsymbol{x} - \boldsymbol{x}'\|$ who converges to a constant quickly as the dimension increases (Liu et al., 2020). Such kind of locality in terms of stationary and monotonic properties result in that they can not reveal more important information in the feature spaces, which largely restricts the performance of kernel methods in complex tasks (Xue et al., 2019). The RF models overcome this issue with the induce of the coefficients $\boldsymbol{\theta}$ and its associated spectral distribution. In specific, the RF model learns a kernel function based on the fixed activation function $\sigma(\cdot)$ indexed by (approximately) infinite random parameters from a prespecified distribution. In terms of the algorithm and implementation, the RF model improves the quality of approximation and reduces the requirement on time and space compared with traditional kernel approximation methods (Liu et al., 2020). This is because that the RF model is able to map features into a new space where the dot product can approximate the kernel accurately, thus improving the quality of the approximation (Yu et al., 2016). Comparing to other kernel methods that mapping $\boldsymbol{x}$ to a high dimensional space, RF uses a randomized feature map to map $\boldsymbol{x}$ to a low-dimensional Euclidean inner product space. Consequently, we can simply use linear learning methods to approximate the result of the nonlinear kernel machine (Rahimi & Recht, 2007), which saves computation time and reduces computation complexity. Also, unlike Nystrom methods or other data dependent methods, RF is a typical data-independent method with an explicit feature mapping. Data-independent implies that RF does not need large samples to guarantee its approximation property(Liu et al., 2020). However, it still fails to provide satisfactory performance for complex tasks due to its representing of a simple stationary kernel only. In contrast, sampling $\boldsymbol{\theta}$ from a mixture distribution would bring in extra computational complexity (Avron et al., 2017). On the other hand, recently, some work have been done via the kernel Neural Network (KDL), a combination of kernel methods and Neural Network, to overcome the limitation of the locality (Xue et al., 2019), and adopt the kernel trick to make computation tractable. In particular, KDL methods incorporate Neural Network methods to kernel functions, i.e., $k(g(\boldsymbol{x}, \boldsymbol{\theta}), g(\boldsymbol{x}', \boldsymbol{\theta}))$, where $g(\boldsymbol{x}, \boldsymbol{\theta})$ is a non-linear mapping given by a deep architecture. KDL trains a deep architecture $g(\cdot; \boldsymbol{\theta})$ indexed by finitely many fixed parameters and then plugs it into a simple kernel function such as a Gaussian kernel. In this way, KDL adaptively estimates basis functions with finitely many parameters at the price of requiring lots of hand tuning work (lack of a principled framework to guide parameter choices), and thus a large number of data size is needed. In this paper, following a similar spirit of KDL, we develop a novel supervised RF method (SRF) to overcome the local kernel's limitation by first adaptively estimating basis functions through (approximately) infinite tuning-free kernel techniques based on low-dimensional variables in the form of $\langle \boldsymbol{x}, \boldsymbol{\theta} \rangle$ with $\boldsymbol{\theta}$ from a simple distribution, and then adaptively estimating the corresponding weights and the unknown link in a supervised way. Most importantly, with the incorporation of the information from the outcome to learn the basis functions, the proposed SRF has excellent predictive performance with the limited data size, in addition to the advantage of being interpretable, and hand-tuning free. It is worth noting that standard RF only has one single layer, which may not thoroughly express the complexity of the data. Instead, SRF includes two layers, which makes it have stronger ability of expression. Moreover, unlike KDL, which only introduces the information of $\boldsymbol{y}$ at the last layer, SRF incorporates the information of $\boldsymbol{y}$ at each layer, leading to a higher prediction power without abundant layers. This idea is very similar to the idea of Conditional Variational Autoencoders(CVAE), which is also known for good performance on limited data size and being energetic efficient (Kingma & Welling, 2013; Sohn et al., 2015). Energetic efficiency is an important aspect of the SRF approach. Compared to CVAE, the proposed SRF method enjoys easier interpretation benefit via its flexible semi-parametric structure.

The proposed SRF has the following contributions: First, computational simplicity. Conventional RF models including training the random features in the implicit kernel learning (Li et al., 2019a), choosing random features via kernel alignments in kernel allegement method (Sinha & Duchi, 2016; Cortes et al., 2010), choosing random features by score functions in the kernel polarization method (Shahrampour et al., 2018), among others, require a huge computational burden. Instead, the SRF model generates the random features randomly from a simple pre-specified distribution. In comparison

to single hidden-layer neural networks (NN-1,(Rumelhart et al., 1986)): $f(\mathbf{x}) = \sum_{j=1}^{k} \sigma(\mathbf{w}_j^\top \mathbf{x} + b_j)$, where $k$ is the number of units in the hidden layer. NN-1 requires to estimate $pk$ parameters $\{\mathbf{w}_j\}_{j=1}^{k}$, while RF models estimate $N$ linear coefficients $\{\alpha_j\}_{j=1}^{N}$. The Projection Pursuit Regression ([PPR,(Friedman & Stuetzle, 1981)) combines GAM and NN-1 by estimating nonlinear functions $f_j$ and projected directions $\mathbf{w}_j$ simultaneously, that is, $f(x) = \sum_{j=1}^{k} f_j(\mathbf{w}_j^\top \mathbf{x})$, which requires extensive computations when $p$ and/or $k$ are large. Furthermore, it is known that we usually require large $N$ to obtain a good approximation on the function space. However, when the number of random features, $N$, is large, directly estimating the combination coefficients of supervised random features using the ridge regression equation 2 is computationally burdensome. The proposed SPF divide all random features into $K \ll N$ blocks. For each block, the ridge regression is adopted to obtain initial predictions on the outcome $y$. Then, PPR is used on the low-dimensional (K) predictors to obtain the final prediction. This step further improves the prediction accuracy by adaptively estimating the combination schemes, in addition to save computational time by avoiding directly running large dimensional ridge regression but in a scalable way. Second, model flexibility and automatic calibration(Wilson et al., 2016). Similarly to generalized additive models (GAM,(Hastie, 2017)), i.e, $f(\mathbf{x}) = \sum_{j=1}^{p} f_j(x_j)$, RF models overcome the curse of dimensionality by mapping $p$-dimensional covariates into one dimensional random feature, i.e., $\langle \boldsymbol{\theta}, \mathbf{x} \rangle$. Different from GAM, the RF model has the capacity to model interactions between covariates using the projected direction $\boldsymbol{\theta}$. The proposed SRF estimates the activation functions in a supervised way for each random feature, which avoids any subjective pre-specified fixed kernel space. It adaptively estimate each function and thus allows different function spaces on each random feature. Therefore, the proposed SRF allows a more complex function space on the variables $\mathbf{x}$ without knowing the true space they belong to. Consequently, the proposed SRF model has a more stable prediction errors in comparison to conventional RF models. Third, model simplicity. Different from multi-layer neural network, the SRF model needs two layers only to achieve good prediction accuracy. As described in the following section, the first layer is a 'nonparametric' random feature through the nonparametric regression method, and the second layer is the projection pursuit, a universal approximator in terms of theoretically approximating any continuous function in $\mathbb{R}^p$ very well, which is extremely useful for regression forecasting due to its semi-parametric structure. More importantly, the estimation of these two layers can be easily solved by using common statistical methods without the need for extensive manual tuning from the users. Finally, model interpretability. As is known to all, neural network is lack of a principled framework to guide choosing parameters, such as architecture, activation functions or optimizers (Wilson et al., 2016). This, combining with unidentification of parameters, leads to the uninterpretability of neural network. Fortunately, our SRF model enjoys good interpretability to some extent. For instance, as we mentioned before, RF model uses linear learning methods to replace nonlinear kernel methods. The biggest advantage of linear learning methods is its interpretability in terms of the coefficient. Significant coefficients implies important directions $\langle \boldsymbol{\theta}, \boldsymbol{x} \rangle$ (Liu et al., 2020), which facilitates the interpretation and understanding the underlying important features.

The rest of the paper is organized as follows. Section 2 introduces proposed SRF with details and algorithm. Section 3 compares the proposed SRF method with other statistical methods under various types of simulated data. Section 4 considers five RWD(Real World Data) examples to evaluate the performance of the proposed SRF method. In Section 5, we summarize this paper with concluding remarks.

## 2 SUPERVISED RANDOM FEATURE

Consider the problem:

$$Y_i = f_0(\mathbf{x}_i) + \varepsilon_i, \tag{3}$$

where $\mathbf{x}_i \in \mathbb{R}^p$ is a $p$-dimensional vector, and the function $f_0$ is unknown. The random errors $\varepsilon_i, 1 \le i \le n$, are independent of each other and of $\mathbf{x}_i$. Assume $\mathbb{E}(\varepsilon_i) = 0$ and $\mathbb{E}(\varepsilon_i^2) = \sigma^2 < \infty$. When the dimension $p$ is larger than 3, the curse of dimensionality in the nonparametric regression occurs. We now introduce the proposed supervised random feature model, denoted as SRF. Firstly, for each random feature, $\langle \boldsymbol{\theta}_j, \mathbf{x} \rangle$, we calculate its prediction on the outcome $Y$. That is,

$$Y_i = f_j(\langle \boldsymbol{\theta}_j, \mathbf{x}_i \rangle) + \varepsilon_i, \quad i = 1, \cdots, n, \tag{4}$$

where $f_j(\cdot)$ is an unknown univariate non-parametric function. Denote its estimator as $\hat{f}_j$, an initial prediction, which can be obtained easily using any nonparametric tools, such as K-NearestNeighbor(KNN), and Kernel density estimation or Kernel regression from Python package statsmodels.nonparametric. It is worthy of pointing out that for each RF $\langle \boldsymbol{\theta}_j, \mathbf{x} \rangle$, we estimate the activating function in a supervised way. By doing this way, we avoid the misspecification issue on the kernel space. Second, the adaptive way on the kernel space relaxes the restriction on the distribution of the random index parameter $\boldsymbol{\theta}_j$. In other words, we can simply sampling $\boldsymbol{\theta}_j$ from a unit ball, and then adaptively estimate the corresponding activating function with the outcome information incorporated. Third, for each RF, the underlying kernel space may not be the same. Thus, with each function estimated independently, we actually obtain a multi-kernel mixed space, which largely improve the prediction power compared to the single-kernel space especially for complex task. Secondly, we further refine the prediction in an aggregated way by minimizing the following ridge-type objective similar to conventional RF models:

$$\frac{1}{n} \sum_{i=1}^{n} \left( Y_i - \sum_{j=1}^{N} \alpha_j \hat{f}_j(\langle \boldsymbol{\theta}_j, \mathbf{x}_i \rangle) \right)^2 + \frac{N\lambda}{p} \|\boldsymbol{\alpha}\|_2^2. \tag{5}$$

Denote the prediction as

$$\hat{f}_{SRF-I} := \sum_{j=1}^{N} \hat{\alpha}_j \hat{f}_j(\langle \boldsymbol{\theta}_j, \mathbf{x}_i \rangle).$$

By treating each initial prediction $\hat{f}_j$ as a candidate model, the SRF method shares similar idea as the stacking methods in model averaging literature (Yao et al., 2018). That is, we aggregate each predictions $\hat{f}_j$ through the weights $\alpha_j$, obtained by minimizing a least-square type criterion. Clearly, weights $\alpha_j$'s could be positive or negative. Instead of all positive weights in conventional modeling averaging methods, allowing positive and negative weights improves prediction power especially when candidate models do not cover the underlying true model (Arce, 1998). Different from stacking methods, SRF involves random features $\boldsymbol{\theta}_j$ leading to a clear identification on important features whose corresponding coefficient $\alpha_j$ is usually large, as show in Simulation-Part three.

To avoid the computation complexity with large $N$, we further divide $N$ into $K$ blocks with equal dimension in each without loss of generality. Then within each block, we run equation 5 to obtain raw predictions $\hat{f}_k^{(1)}(\mathbf{x}_i) = \sum_{j=1}^{N_k} \hat{\alpha}_j^k \hat{f}_j(\langle \boldsymbol{\theta}_j, \mathbf{x}_i \rangle)$. Base on $K$ predictors, $\hat{f}_k^{(1)}(\cdot), k = 1, \cdots, K$, we obtain a further refined prediction by minimizing the following objective

$$\frac{1}{n} \sum_{i=1}^{n} \left[ Y_i - g \left( \sum_{k=1}^{K} \beta_k^{(1)} \hat{f}_k^{(1)}(\mathbf{x}_i) \right) \right]^2, \tag{6}$$

where $g$ is an unknown nonparametric function. This step further improves prediction accuracy through the non-parametric aggregation link function $g$ and additional weight parameters $\beta_k$. Specifically, the non-parametric aggregation link function $g$ extracts interaction information for each features, and the product term $\alpha_j^k \times \beta_k$ enjoys the ability of extracting hierarchical information from each feature, which is similar to the two-layer NN. Different from the two-layer NN, who prespecifies an activation function and the final link function, the proposed SRF estimates each activating function $f_j$ and the final link function $g$ with the outcome information incorporated. Thus, it has higher prediction power due to the use of multi-kernel mixed space and flexible interaction expression with the non-parametric form, in a supervised way. The estimator can be obtained by PPR. The final prediction is defined as

$$\hat{f}_{SRF-II} = \hat{g} \left( \sum_{k=1}^{K} \hat{\beta}_k^{(1)} \hat{f}_k^{(1)}(\mathbf{x}_i) \right).$$

The entire procedure is given in the following Algorithm 1.

---

**Algorithm 1** Algorithm for SRF-II

1: SRF$\{y_i, \mathbf{x}_i\}_{i=1}^n$, $N$, $K$ {Input}
2: Randomly generate $N$ directions $\boldsymbol{\theta}_j, j = 1, \cdots, N$: $\boldsymbol{\theta}_j \sim \text{Unif}(\mathbb{S}^{p-1}(\sqrt{p}))$.
3: Obtain supervised random features $\hat{f}_j, j = 1, \cdots, N$ using equation 4.
4: Obtain initial $K$ raw esitmators $\hat{f}_k^{(1)}, k = 1, \cdots, K$ by minimizing equation 5.
5: Obtain $\hat{f}(\mathbf{x}) = \hat{g}\left(\sum_{k=1}^K \hat{\beta}_k^{(1)} \hat{f}_k^{(1)}(\mathbf{x})\right)$ by minimizing the objective equation 6.
6: **return** $\hat{f}$. {Output}

---

## 3 SIMULATION STUDIES

This section evaluates the performance of the proposed SRF method based on various types of simulated data. We compare the prediction results with other statistical methods including Basic Random Feature Regression (Relu-I), One Layer Kernel Regression (SRF-I), Advanced Basic Random Feature Regression (Relu-II), Two Layers Kernel Regression - Projection Pursuit (SRF-II), Random Forest, One Layer Neural Network (NN-1) and Two Layers Neural Network (NN-2). We consider four settings for the regression function $f_0(\cdot)$: (a) Linear: $f_0(\boldsymbol{X}) = 2X_1 + X_2 + 3X_3$ ; (b) Composite: $f_0(\boldsymbol{X}) = \cos\left\{X_1 + \cos(X_1) + X_2^2 + e^{X_2/3} + X_5X_3X_4 + \cos(X_5) + 2X_6 + X_7^2 + X_8X_9 + X_{10}\right\}$; (c) Nonlinear: $f_0(\boldsymbol{X}) = (X_1 + X_2 + X_3)^2 + 1$; (d) More complex: $f_0(\boldsymbol{X}) = \cos\left\{X_1 + 2X_1X_2 + X_3^2 + \sin(X_4) + \exp(X_5)\right\} + \exp(X_6 + X_7) + \cos(X_8 + X_9 + X_{10}^2) + \sin(X_1 + X_5)^2$. Here, $X_i$ represents the ith dimension of $\boldsymbol{X} \in \mathbb{R}^p$.

Under all settings, we generate $n = 300$ data with $p = 100$ covariates. We generate covariates $\boldsymbol{X}$ from a multivariate normal distribution, $N(\boldsymbol{0}, \Sigma)$, with three correlation structures: (I) Independence, i.e., $\Sigma = \boldsymbol{I}$. (II) Fixed correlation structure, i.e., all off-diagonal component of $\Sigma$ equals to $0.5$. (III) Random correlation structure, that is, each off-diagonal component of $\Sigma$ is randomly generated from a uniform distribution $\text{Unif}(-1, 1)$. The random error $\epsilon$ in the regression function $Y = f_0(\boldsymbol{X}) + \epsilon$, is generated from a normal distribution, i.e., $\varepsilon \sim N(0, 0.1)$. We replicate each simulation 100 times.

To determine the number of random features $N$, extensive simulation results show that when $N = 12000$ and $N = 24000$ for independent and correlated covariates, respectively, the prediction accuracy keeps steady and larger N does not bring significant improvement, as show in the Figure 4 (See Appendix A). Thus, for computational simplicity, we take $N = 12000$ and $N = 24000$ for simplicity. The larger N required for the correlated covariates is understandable, since randomly generated RF's may be correlated and share similar information due to the correlation among covariates leading to a larger $N$ to thoroughly capture covariates information. The regularization parameter $\lambda$ is determined by the model complexity. Basically, in order to ensure the stability of the model and the accuracy of the estimation, larger $\lambda$ is used as the model complexity increased. We compare the prediction performance in terms of predicted mean square error (MSE) ,i.e $\frac{1}{n}\sum_{i=1}^n (\hat{y}_i - y_i)^2$ and Scaled MSE, i.e $\sqrt{\frac{1}{n}\sum_{i=1}^n (\frac{\hat{y}_i - y_i}{y_i/2})^2}$ , with additional data with size 100. To ensure visualization of simulation results, we exclude $5\% - 10\%$ outliers, and the number 2 in the denominator of Scaled-MSE amplifies the results and is more conducive to observing the difference in visualization.

The results for models (a)-(d) are showed in Table 1 and Figure 1 under independent covariates (I). Particularly, Table 1 summarizes the average of the Scaled-MSE and MSE based on 100 replications, and Figure 1 shows the box-plot for the Scaled-MSE. Based on Table 1 and Figure 1, we can see that when the model is simple, as linear model in scenario (a), two-layer type methods does not have better prediction accuracy than their corresponding one-layer counterparts. Interestingly, Scaled-MSE and MSE have different preferences on models. SRF-I, SRF-II and NN-1 have smaller Scaled-MSEs, while Relu-II and Random Forest have smaller MSEs. The reason for this problem may be that the linear model is too simple, and kernel regression and projection pursuit are prone to over-fitting. SRF shows its advantage as the model complexity increased. For models (b) and (c), SRF-I and SRF-II have better performance than others in terms of both Scaled-MSE and MSE. Also, because of the complexity of the composite function, all the two layers models perform better than their corresponding one layer counterparts. However, Neural Network do not show their advantages under models (b) and (c). Further, for more complex data as model (d), Neural Network works well. SRF-I

and SRF-II defeat Relu-I and Relu-II in terms of Scaled-MSE. Random Forest still remains stable. It is worth noting that under this scenario, the two-layer models have significantly better and more stable prediction performance than one-layer models besides Neural Network.

Table 1: Scaled-MSEs and MSEs for models (a)-(d) of various methods including Relu-I, SRF-I, Relu-II, SRF-II, Random Forest, NN-1 and NN-2. Two-layer type methods have no obvious advantages for linear models. As model complexity increases, for composite and non-linear models, SRF-I and SRF-II perform better than others for both Scaled-MSE and MSE. Further, for more complex model , SRF-I and SRF-II work well on Scaled-MSE. NN-1 and NN-2 has comparable performance under model (d) setting.

| Model | Linear | | Composite | | Non-Linear | | More Complex | |
|---|---|---|---|---|---|---|---|---|
| | Scaled MSE | MSE | Scaled MSE | MSE | Scaled MSE | MSE | Scaled MSE | MSE |
| Relu-I | 16.75 | 3.20 | 13.37 | 0.57 | 3.78 | 19.56 | 2.33 | 20.11 |
| SRF-I | 5.22 | 12.89 | 4.97 | 0.49 | 2.61 | **17.18** | 1.73 | 19.07 |
| Relu-II | 7.28 | 2.77 | 8.66 | 0.57 | 3.53 | 19.09 | 3.81 | 20.80 |
| SRF-II | 6.35 | 13.70 | **3.75** | **0.48** | **1.65** | 17.59 | **1.04** | 21.84 |
| Random Forest | 9.20 | **1.82** | 6.02 | 0.51 | 3.11 | 15.60 | 1.15 | **10.87** |
| NN-1 | **4.50** | 6.52 | 15.21 | 0.85 | 4.49 | 23.64 | 1.51 | 33.64 |
| NN-2 | 7.93 | 8.16 | 8.08 | 0.56 | 4.53 | 27.14 | 1.53 | 22.01 |

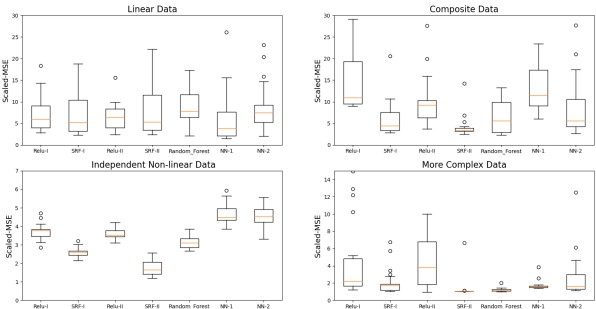

Figure 1: Box-plot of Scaled-MSEs for models (a)-(d) under independent covariates (I). Clearly, except under linear model setting (a), SRF-II has smallest Scaled-MSEs under all other models.

To show the effects of various correlation among covariates, we further compare the results under model (c) for both random and fixed covariance matrices (II) and (III). The results are shown in Figure 5 (See Appendix B) . Obviously, SRF-II still has the best performance. Compared to Figure 1 (independent scenario), both Scaled-MSEs and MSEs are larger because of the effects of various correlation. Except that, all the models have similar performance under correlated covariates.

In order to show the interpretability, we consider the following two criterions under model (c). For the first criterion, we calculate the difference of sum of absolute weights $\omega$ on the three important covariates between the maximum absolute $\alpha_{\max} = \max_j |\alpha_j|$ and the minimum absolute $\alpha_{min} = \min_j |\alpha_j|$. Ideally, a large absolute value of $\alpha_j$ implies the more importance of the feature. Thus, we compare the proportion of the differences larger than 0 out of 50 replicates for 20 $\alpha$'s at a time, termed as Maxmin. For comparison, we also compare the difference between randomly chosen $\alpha_j$ and $\alpha_{\min}$, termed as Ranmin. For the second criterion, we compare the significant $\omega$ elements (the first three elements) with the non-significant $\omega$ elements. For comparison, we consider three non-significant $\omega$ at prespecified fixed positions (Fixpos) or randomly chosen positions (Ranpos). Also, we compare the difference between randomly chosen three covariates and randomly chosen covariates (Ranran). Similarly, we calculate the proportions of the differences larger than 0 out of 50 replications for 20 $\alpha$'s at a time. The results are show in Figure 2. From Figure2 we can see that, Ranpos and Fixpos have significantly larger proportions, up to $56\%$ improvement on average, than Ranran, indicating that important features do have larger values on $\omega$. Maxmin also has significantly larger proportion than Ranmin, up to $80\%$ improvement on average, implying that larger $\alpha_j$ does represent an important direction. Therefore, SRF-II is meaningful to determine important directions with large value of $\alpha_j$.

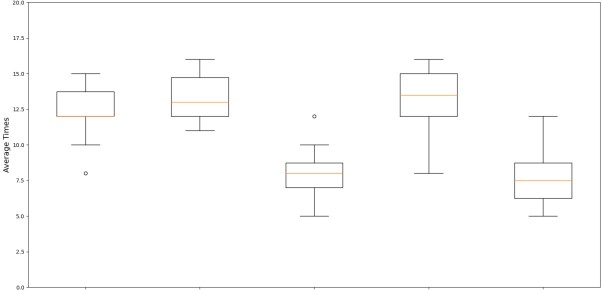

Figure 2: The average times of differences larger than 0 out of 50 replicates under five different indexes, Maxmin, Ranmin, Fixpos, Ranpos, and Ranran, respectively. Compared to Ranran and Ranmin, Larger times in Maxmin, Fixpos and Ranpos show that larger $\alpha$ values are related to important features.

## 4 REAL DATA EXAMPLES

**List of datasets.** Our real data experiments consider the following datasets. All the datasets are publicly available. More details about these datasets, including the size of the data and the number of features, are provided in Table 2.

- **Abalone** was collected form UCI (University of California-Irvine) Machine Learning Repository with data size n= 800. Its objective is to predict the age of abalone (Number of rings) based on individual abalone measurements. It contains seven features including length, diameter, height, whole weight, shucked weight, viscera weight, and shell weight.

- **Boston** was collected from Sklearn Machine Learning Repository with data size n = 478. This data set is about Boston House Prices and is one of the most famous regression task datasets. It contains thirteen features, which are CRIM, ZN, INDUS, CHAS, NOX, RM, AGE, DIS, RAD, TAX, PTRATIO, B and LSTAT(See Appendix C for details). The objective is to predict how those aforementioned features affected the house price, MEDV(Median value of owner-occupied homes in $ 1000's ).

- **Wine** was also collected from UCI Machine Learning Repository with data size n =1000. The white wine quality data set contains eleven features, which are fixed acidity, volatile acidity, citric acid, residual sugar, chlorides, free sulfur dioxide, total sulfur dioxide, density, PH value, sulphates, and alcohol. These eleven independent variables are used to predict the quality(based on sensory data) of each white wine.

- **Auto MPG** was collected from Kaggle.com. This MPG data is about n = 393 automobile fuel consumptions in miles per gallon with three multi-valued discrete attributes and five continuous attributes. These eight attributes are MPG(miles per gallon), number of engine cylinders, engine displacement, horsepower, vehicle weight, acceleration, model year, and origin. The objective is to predict MPG based on the other seven features.

- **Song Popularity** was collected from Kaggle.com. Recently, there has been increasing research work into the relationship between the popularity of a song and its certain factors. The main goal is to predict a song's popularity based on several factors. In this dataset, thirteen factors including song duration, acousticness(electronic music or not, 0 to 1), danceability, energy, instrumentalness(pure music or not, 0 to 1), key, liveness, loudness, audio mode, speechiness, tempo, time signature and audio valence(positive or negative psychological feelings, 0 to 1), were collected. For ease of illustration, the data with key 4 is considered leading to n= 1307 data size.

In the data preprocesses, min-max normalization for each continuous variable is applied for all datasets except Auto MPG and Song Popularity, in which, z-score normalization is used.

**Prediction results.** All predicted MSEs and Scaled-MSEs (with standard deviation) are reported in Table 2. It can be seen from Table 2 that, SRF-II has the best performance in terms of both Scaled-MSEs and MSEs, with the reduction in Scaled-MSEs and MSEs compared to Relu-I ranges from $9 - 34\%$ and $2 - 28\%$, respectively, for all datasets. With limited data size, NN-2 and NN-1 have instability issues. Particularly, for dataset Abalone, conventional RF models and the proposed SRF have comparable performance, and two-layer models do not show there advantages compared with one-layer counterparts. This is probably due to the underlying simple structure of the data. For the other four datasets, two-layer models have smaller prediction errors than their corresponding one-layer counterparts.

Table 2: Prediction results including Scaled-MSEs (with standard deviation) and MSEs for all five datasets as well as data size, number of features, and publicly available or not. In data Abalone, Proposed SRF methods do not show obvious advantage. For the other four datasets, two-layer models defeat their corresponding one-layer counterparts. The proposed SRF performs best on the dataset Boston,Auto MPG, and Song Popularity. For all the five datasets, the proposed SRF-II works particularly well. ($\star$: Due to the extreme instability of two-layers Neural Network, part of the error data was deleted. )

| Dataset | Abalone | | Boston | | Wine | | MPG | | Song | |
|---|---|---|---|---|---|---|---|---|---|---|
| Data size | 800 | | 478 | | 1000 | | 393 | | 1307 | |
| Number of features | 7 | | 13 | | 11 | | 8 | | 12 | |
| Available (YES/No) | YES | | YES | | YES | | YES | | YES | |
| Results | Scaled MSE | MSE | Scaled MSE | MSE | Scaled MSE | MSE | Scaled MSE | MSE | Scaled MSE | MSE |
| Relu-I | $0.48 \pm 0.032$ | **6.29** | $0.62 \pm 0.030$ | 18.94 | $0.39 \pm 0.017$ | 0.93 | $2.63 \pm 0.372$ | 0.23 | $6.62 \pm 1.020$ | 1.12 |
| SRF-I | **$0.44 \pm 0.050$** | 6.57 | $0.60 \pm 0.038$ | 14.25 | $0.37 \pm 0.011$ | 0.96 | $2.42 \pm 0.372$ | **0.20** | $6.01 \pm 1.049$ | 1.11 |
| Relu-II | **$0.44 \pm 0.016$** | 6.82 | $0.48 \pm 0.020$ | 14.18 | $0.36 \pm 0.008$ | 0.98 | $2.21 \pm 0.550$ | 0.22 | $4.70 \pm 0.615$ | 1.07 |
| SRF-II | **$0.44 \pm 0.020$** | 6.32 | **$0.47 \pm 0.026$** | **13.55** | $0.35 \pm 0.011$ | 0.89 | **$1.73 \pm 0.608$** | 0.25 | **$4.35 \pm 0.615$** | 1.09 |
| Random Forest | $0.67 \pm 0.001$ | 12.18 | $0.79 \pm 0.001$ | 34.62 | **$0.33 \pm 0.001$** | **0.85** | $1.97 \pm 0.001$ | 0.24 | $5.84 \pm 0.001$ | **0.64** |
| NN-1 $\star$ | $0.50 \pm 0.007$ | 7.62 | $0.61 \pm 0.003$ | 19.77 | $0.39 \pm 0.002$ | 0.98 | $3.29 \pm 0.026$ | 0.22 | $5.37 \pm 0.677$ | 0.99 |
| NN-2 $\star$ | $0.51 \pm 0.018$ | 7.93 | $0.57 \pm 0.040$ | 24.86 | $0.36 \pm 0.005$ | 0.88 | $3.31 \pm 0.238$ | 0.28 | $6.89 \pm 0.043$ | 0.95 |

**Interpretability results.** In the previous simulation section, we have confirmed that SRF-II is meaningful to determine important directions with large value of $\alpha_j$. Thus, for the real data examples, we identify the significant variables according to the magnitude of the absolute values of $\omega$. Particularly, we identify significant variables by integrating a ranking of 50 times for every dataset. The proportion of each variable in the first few ranks (depending on the number of features) in each dataset is reported in Figure 3.

- **Abalone** Due to the characteristic of the dataset itself, there are no significant and non-significant variables, which means that each variable has a certain effect on abalone age.

- **Boston** RM,LSTAT and B are three significant variables in this dataset. The significance of INDUS and AGE were moderate. According to the analysis from github.com, LSTAT and RM have the biggest correlation coefficient with MEDV. Persons tend to have a lower proportion of low status people around their houses, and more rooms imply a bigger house. Interestingly, we found that B is also an important factor for MEDV unlike what others have found. Other variables have less obvious effects on MEDV.

- **Wine** In this dataset, alcohol, volatile acidity, and chlorides are relatively significant compared to the others. Because of the nature of wine, the best quality is achieved by a balance of all variables, so there are no particularly significant variables in this dataset. First of all, it is often said that the higher the alcohol content, the better the wine. Then, too much volatile acidity can cause the wine to smell pungent. At last, the right amount of chloride can extend the life of a wine, but too much can produce an unpleasant taste.

- **Auto MPG** For this dataset, weight and displacement have obvious significant characteristic, and year and horsepower are kind of significant. Other variables are less likely to affect the value of MPG. Based on other people's correlation analysis of this dataset on Kaggle.com, the largest absolute correlation coefficients with MPG are weight and displacement. This conclusion is consistent with our results. Through common sense, we clearly know that the heavier the car, the bigger the MPG, and then the bigger the displacement, the bigger the MPG. It is worth mentioning that the year will also have an effect on the value of MPG, because the higher the year, the more serious the aging of auto parts, which will lead to the increase of MPG.The other variables are not significant because their correlation coefficients with weight, displacement and horsepower are too high.

- **Song Popularity** Our results are not exactly the same as other people's correlations conclusion on Kaggle.com, but they are roughly the same. Audio valence and loudness are two significant variables in this dataset. Also, acousticness, danceability and instrumentalness are kind of significant. Everyone loves to listen to songs with positive psychological feeling, so it is easy to understand that audio valence has a great influence on song popularity. And, few people like loud songs, so loudness is also a significant variable. Electronic music, dance music and pure music have unique audiences, so they have some influence on song popularity. However, for other variables, such as liveness or tempo, the audience doesn't pay much attention to those. Therefore, they are non-significant variables.

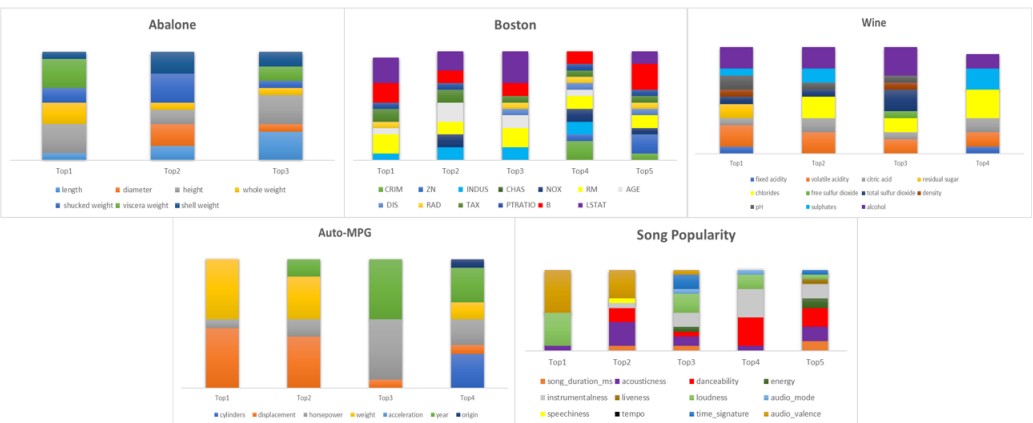

Figure 3: The proportion of each variable in the first few ranks (depending on the number of features) in each dataset is shown. A Longer color bar in the histogram represents a higher proportion, implying the significant level of the feature. It's easy to see that Abalone and Wine don't have obviously significant variables. For Boston data, RM,LSTAT and B are three significant variables. For Auto MPG, weight and displacement have obvious significant characteristic. Audio valence and loudness are two significant variables in the data Song Popularity.

## 5 CONCLUDING REMARKS

This paper proposed a novel and computational efficient method for general nonparametric problems. To the best of our knowledge, we are the first to propose combining the advantages of random features and neural networks to come up with a new feed-forward two-layer nonparametric estimation method. Extensive simulation results and experimental data show that SRF-II has excellent prediction performance and good interpretation. More specifically, the proposed SRF-II improved the prediction error compared to Relu-I ranges from $9 - 34\%$ and $2 - 28\%$ across five datasets. More importantly, SRF-II performs well with limited data size and thus is energetic efficient. However, there are still three limitations in this paper. At first, for computational simplicity, we take N = 12000 and N = 24000 in these cases. However, optimal choices or clear criterions are not clear yet. Secondly, in this paper, we only consider the regression problem, not the classification problem. Classification problems based on RF require extra computational burden due to the non linear structure compared to least squares. Nevertheless, extending SRF-II to classification problems, especially for image classification problems, are very meaningful. In the end, we did not yet consider variable selection methods to generate a more simplified model, which well deserved further studies to improve the prediction accuracy and computational efficiency further.

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

## A   APPENDIX

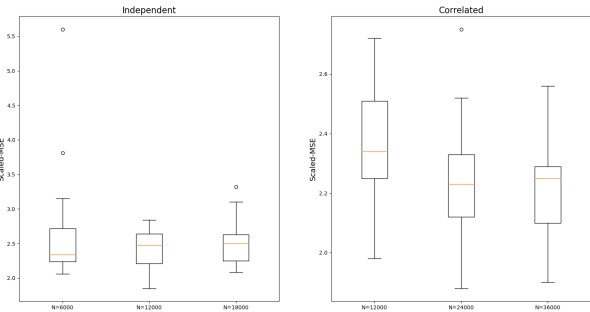

Figure 4: The box figures of scaled predicted mean square error under model (c) for independent covariates generated from (I) with $N = 6000, 12000, 18000$ and correlated covariates generated from (III) with $N = 12000, 24000, 36000$.

## B   APPENDIX

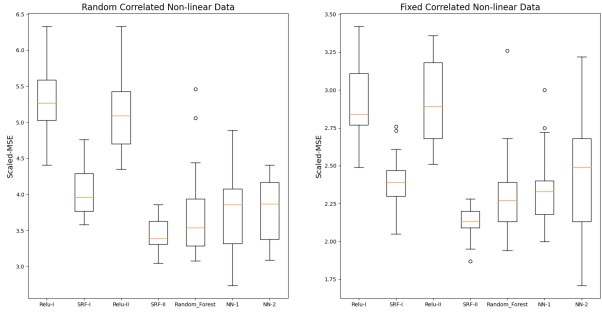

Figure 5: Box figures for Random and Fixed correlated Non-linear Data. SRF-II continues to perform well in both cases. Both Scaled-MSEs and MSEs are larger because of the correlation of covariates.

## C   APPENDIX

Boston House Prices Dataset Attribute Information (in order):

- CRIM     per capita crime rate by town
- ZN     proportion of residential land zoned for lots over 25,000 sq.ft.
- INDUS     proportion of non-retail business acres per town
- CHAS     Charles River dummy variable (= 1 if tract bounds river; 0 otherwise)
- NOX     nitric oxides concentration (parts per 10 million)
- RM     average number of rooms per dwelling
- AGE     proportion of owner-occupied units built prior to 1940
- DIS     weighted distances to five Boston employment centres
- RAD     index of accessibility to radial highways
- TAX     full-value property-tax rate per \$10,000
- PTRATIO     pupil-teacher ratio by town
- B     $1000(Bk - 0.63)^2$ where Bk is the proportion of blacks by town
- LSTAT     % lower status of the population

- MEDV                           Median value of owner-occupied homes in $ 1000's

