# OpenReview forum: "Supervised Random Feature Regression via Projection Pursuit"
_ICLR.cc/2023/Conference — Submitted to ICLR 2023_

### Official Review · Reviewer_ZaoF · 2022-10-18

**Confidence:** 4
**Correctness:** 2
**Technical Novelty And Significance:** 1
**Empirical Novelty And Significance:** 2
**Recommendation:** 3

**Clarity, Quality, Novelty And Reproducibility:**

**Clarity**: The presentation in this paper can be clearly improved.
**Quality and Novelty**: The theoretical and empirical contributions in the paper are of limited significance and novelty. Please see my detailed comments below.
**Reproducibility**: good.

**Strength And Weaknesses:**

**Strength**: I do not see very strong points in the paper, from either a theoretical or empirical viewpoint.
** Weaknesses**: There is (almost) no theory in the paper, and the baselines compared in the experiments are weak. The presentation of the paper is, in general, poor.

**Summary Of The Paper:**

In this paper, the authors proposed a supervised random features (SRF) regression method that combines the ideas of RF kernel learning and of (simple) neural networks (NNs) model.
The authors claimed that the proposed SRF approach improves a few previous efforts in that it yields better performance with a relatively small amount of data (when compared with, e.g., kernel neural networks), and can be computed very efficiently (when compared with, e.g., implicit kernel learning or kernel alignment methods mentioned at the end of the second page). However, none of these claims are well-supported by solid and rigorous theory.
Some limited experiments were provided on a few datasets and a few simple models to illustrate the advantages of the proposed SRF approach.


**Summary Of The Review:**

As I mentioned above, the theoretical and empirical contributions in the paper are of limited significance and novelty, and the presentation in the paper can be clearly improved. Please see my detailed comments as follows:
* P1, abstract: my personal, and possibly naive, understanding of the neural network (NN) model is that it is parameterized by a sequence of weight matrices and bias terms, and therefore a parametric model in some sense.
* P1, introduction: "reduce the extensive hand tuning form the user for training": I get confused by this sentence, does this mean classical kernel learning needs a lot of hand tuning and that can be avoided by applying RF techniques?
* Please use $\max$ instead of max, $\sin$, and $\cos$ for sin and cos activation functions, respectively.
* P2: The author claimed that this contribution improves kernel neural networks (KDL), which need a huge number of data, by proposing a novel supervised RF approach that works well even with a limited amount of training data. This advantage, however, is only evaluated empirically, no solid theoretical arguments are provided in the paper.
* P2: Most computational or statistical advantages of the proposed SRF approach are stated without any empirical or theoretical evaluations.
* P2 and P3: the introduction contains extremely long and wordy paragraphs that try to discuss the advantages of the proposed SRF approach, which is hardly readable and not easy to understand. It would be helpful to at least divide them into subsections or paragraphs.
* The figures are hardly visible, which clearly harms the readability of the paper.

---

### Official Review · Reviewer_hYRF · 2022-10-23

**Confidence:** 3
**Correctness:** 3
**Technical Novelty And Significance:** 1
**Empirical Novelty And Significance:** 2
**Recommendation:** 1

**Clarity, Quality, Novelty And Reproducibility:**

There are several parts of the paper that in my opinion are not sufficiently detailed to be clear.

The description in page 6 is confusing - the test design here sues prior knowledge of the covariates involved; this is in contrast to methods that evaluate each covariate under some score function and show distinct values for covariates involved vs. others. Similarly, the description in Page 8 does not clearly state how the covariates were found to be significant (e.g., how the "ranking" is performed).

Algorithm 1 step 3 refers to eq. 4, but the equation does not provide a method to obtain supervised random features. It is also not clear how eqs. 5 and 6 is to be minimized in Steps 4 and 5, respectively. These should more clearly describe the minimization procedure (which I assume would describe how optimal coefficients alpha and beta are found).
Similarly a clearer description of the initialization of the estimators for the f_j should be provided (which seems to me would have to restrict to a class of candidates).

Given that the proposed approach is a combination of two well-known methods, the numerical results do not provide clear trends, and there is no analytical contribution, my opinion is that the novelty is scant.

**Strength And Weaknesses:**

The literature on RF models uses a single layer for analytical tractability. This manuscript does not provide any analytical results on the two-layer extension.

The claimed interpretability aspects are not clear to me - the inherent use of random features would obscure the interpretation of the learning process. From the description of the numerical experiment it appears that the approach to interpretability may involve multitudes of RF models, but this does not seem like a comparable approach to that of other methods that rely on a single learned model.

The other claimed contributions are also present in the original RF method.

Table 1 and Figure 1 are not surprising - two-layer models perform better than one when the learning problem is sufficiently difficult.
Figures 1 and 3 should be larger (unreadable text).

Table 2 also does not show a clear distinction between existing methods and the proposed on performance.

Unfortunately it is not clear to me how these weaknesses can be addressed in a revision.

**Summary Of The Paper:**

The paper proposes an adaptation of the random feature method approach for kernel methods akin to a two-layer neural network.

**Summary Of The Review:**

One would expect that a dual-layer extension of RFs would perform better than a single layer version. Furthermore there are no analytical contributions for the extension proposed. Finally, several portions of the paper are not sufficiently detailed to make a clear argument for the claimed new benefit of interpretability.

---

### Official Review · Reviewer_iHFk · 2022-10-27

**Confidence:** 3
**Correctness:** 3
**Technical Novelty And Significance:** 2
**Empirical Novelty And Significance:** 2
**Recommendation:** 3

**Clarity, Quality, Novelty And Reproducibility:**

- **Clarity:** The paper could be made easier to read and could benefit from some proofreading to iron out typos (e.g. Page 2: "This is because that the RF", "Comparing to other kernel methods that mapping x to a high dimensional space [...]" etc.)

- **Originality:** I did not find any idea in this paper particularly original.


**Strength And Weaknesses:**


- **Strength:** While much work has been done on constructing arbitrarily expressive kernels [1, 2] and scaling up kernel-based regression methods, more work is needed to make learning more efficient in kernel methods. Borrowing from deep learning to learn expressive families of kernels, as attempted by this paper, is laudable.


- **Weaknesses:**

*Clarity:* The paper is hard to read and contains too many typos.

*Comparison to Deep Learning:* It is unclear why this approach would perform better than vanilla deep learning. A strong intuition and substantially more experiments are needed to make this case.

*Comparison to expressive kernel:* This paper is missing much of the literature on expressive kernel methods, especially Generalized Spectral Kernels [1]. In particular, [2] introduced kernel families (namely GSKs) that are general-purpose in that they contain kernel that can perform as well as any other kernel not in the family, stationary or non-stationary. Additionally, a flurry of methods have been developed to scale up kernel regression. It would have been interesting to discuss what benefits this approach has over GSKs.

- **Additional Comments:** Page 1: No condition is required for Eq (1) to be Kernel Ridge regression. Basis function regression with Ridge penalty is always kernelized Ridge regression. The kernel implied by Eq (1) is random and the behavior as $N \to \infty$ pertains to the convergence of the random kernel to a deterministic kernel.


[1] Samo, Y.L.K. and Roberts, S., 2015. Generalized spectral kernels. arXiv preprint arXiv:1506.02236.
[2] Samo, Y.L.K., 2017. Advances in kernel methods: towards general-purpose and scalable models (Doctoral dissertation, University of Oxford).

**Summary Of The Paper:**

The paper proposes an approach for boosting the effectiveness of kernelized Ridge regression by first learning a set of random features through a deep-learning inspired preprocessing step, then applying kernelized Ridge regression to the learned features.

**Summary Of The Review:**

The paper should be proofread, and more intuition and more experiments should be added to argue the benefits relative to vanilla deep learning or expressive kernel methods.

---

### Decision · Program_Chairs · 2023-01-20

**Decision:**

Reject

**Justification For Why Not Higher Score:**



**Justification For Why Not Lower Score:**



**Metareview: Summary, Strengths And Weaknesses:**

 Various serious issues were raised with this submission; no author rebuttal arrived.